# People with more extreme attitudes towards science have self-confidence in their understanding of science, even if this is not justified

Cristina Fonseca[1], Jonathan Pettitt[2], Alison Woollard[3], Adam Rutherford[4], Wendy Bickmore[5], Anne Ferguson-Smith[6], Laurence D. Hurst [7,8]*

1 The Genetics Society, London, United Kingdom, 2 School of Medicine, Medical Sciences and Nutrition, University of Aberdeen, Institute of Medical Sciences, Aberdeen, United Kingdom, 3 Biochemistry Department, University of Oxford, Oxford, United Kingdom, 4 Department of Genetics, Evolution and Environment, University College London, London, United Kingdom, 5 MRC Human Genetics Unit, Institute of Genetics and Cancer, University of Edinburgh, Edinburgh, United Kingdom, 6 Department of Genetics, University of Cambridge, Cambridge, United Kingdom, 7 The Milner Centre for Evolution, Department and Biology and Biochemistry, University of Bath, Bath, United Kingdom, 8 Wissenshaftskolleg zu Berlin, Berlin, Germany

* l.d.hurst@bath.ac.uk

**Data Availability Statement:** All scripts and data are available at doi: 10.5281/zenodo.7289133.

## Abstract

People differ greatly in their attitudes towards well-evidenced science. What characterises this variation? Here, we consider this issue in the context of genetics and allied sciences. While most prior research has focused on the relationship between attitude to science and what people know about it, recent evidence suggests that individuals with strongly negative attitudes towards specific genetic technologies (genetic modification (GM) technology and vaccines) commonly do not objectively understand the science, but, importantly, believe that they do. Here, using data from a probability survey of United Kingdom adults, we extend this prior work in 2 regards. First, we ask whether people with more extreme attitudes, be they positive or negative, are more likely to believe that they understand the science. Second, as negativity to genetics is commonly framed around issues particular to specific technologies, we ask whether attitudinal trends are contingent on specification of technology. We find (1) that individuals with strongly positive or negative attitudes towards genetics more strongly believe that they well understand the science; but (2) only for those most positive to the science is this self-confidence warranted; and (3) these effects are not contingent on specification of any particular technologies. These results suggest a potentially general model to explain why people differ in their degree of acceptance or rejection of science, this being that the more someone believes they understand the science, the more confident they will be in their acceptance or rejection of it. While there are more technology nonspecific opponents who also oppose GM technology than expected by chance, most GM opponents fit a different demographic. For the most part, opposition to GM appears not to reflect a smokescreen concealing a broader underlying negativity.

**Funding:** The work was enabled by funding from The Genetics Society to the Chair of their Public Engagement committee (AW). No grant number specified. The funders had no role in study design, data collection and analysis, decision to publish, or preparation of the manuscript.

**Competing interests:** The authors have declared that no competing interests exist.

**Abbreviations:** GM, genetic modifications; MRS, Market Research Society; OSD, objective–subjective deficit; PUS, Public Understanding of Science; SAU, subjective science understanding.

## Introduction

In an age where scepticism of well-evidenced science has negative consequences for individuals and society, it is important to understand why people have such discordant attitudes towards well-evidenced science [1,2]. A first step in such an enterprise is to determine what characterises the between-individual variation in attitudinal assessments of science and technology. In the Public Understanding of Science (PUS) literature, much debate has focused on the relationship between factual knowledge of science and attitude positivity; it is commonly found that lower factual knowledge is associated with more negative attitudes (for meta-analysis see [3]). A mooted mechanistic explanation for the association between negativity of attitude and lack of textbook science knowledge is a fear of the unknown [3]. It has, however, been recently reported that those opposing genetic modification (GM) technology as applied to food [4] and vaccines [5], while having low levels of understanding of the science (objective knowledge), nevertheless report that they do understand the science (subjective understanding). The same has most recently been reported for a diversity of well-evidenced science issues [6], as well as for anti-establishment voting patterns [7]. This accords with earlier evidence that what people think they understand about a subject is related to attitude towards that subject [8,9]. That the overconfident—those whose self-assessed understanding exceeds their factual knowledge—are more prone to negative appraisals of science suggests that a fear, disgust, or distrust of what they believe to be the case, rather than of the unknown, underpins their attitudinal position.

The precise mechanism(s) by which science overconfidence might lead to negative attitudes is unclear. Existing research [4,5] has suggested involvement of Dunning–Kruger type effects [10] where the least competent lack also the ability to understand their limitations. However, it is not obvious that Dunning–Kruger effects are either necessary or sufficient as an explanation in this context. They are not sufficient in that, a priori, overconfidence could just as well lead to strong endorsement of a partially understood scientific consensus position. Motta and colleagues [5], following Camargo and Roy [11], address this, arguing that overconfident individuals may be unable to recognise both their own poor comprehension and the expertise of others [5]. In turn, overconfidence could lead to confident dismissal of reliable sources of information allied with an openness to misinformation, thus generating a strongly negative attitude towards the scientific consensus position [5]. We can also envisage circumstances where Dunning–Kruger effects are not a necessary condition. For example, if someone is confident in a false belief that only GM tomatoes contain genes, they might well also believe that the modified genes might transfer to them on consumption, much as a pesticide on a crop might enter their system (for evidence of such coupled thinking, see Discussion). This type of misconception, then, might easily lead to strongly negative appraisal of GM technology in those who are confidently misinformed. Such individuals would be classified as overconfident because their "textbook" science knowledge is weak, but their subjective assessment of their understanding is high. If this is what is happening, then we do not need to evoke any inability to process and logically connect information: the individuals just happen to strongly accept erroneous information and make downstream logical connections.

Here, we do not directly approach, and make no presumptions about, the underlying causality. Rather, our objective here is to extend this prior work [4–6] in 2 previously largely unexplored regards with a view to synthesising a possible general model of attitude to genetical science. First, we ask whether the extremity of attitudinal strength, either positive or negative, is predicted by subjective understanding (i.e., what people think they know and understand). Second, we ask whether trends associated with attitude and knowledge are contingent on specification of technology.

We take as our starting point the fact that in 2 instances, GM foods and MMR vaccine [4,5], those who are most negative in attitude believe they well understand the science but in practice do not, what we refer to as an objective–subjective deficit (OSD). We extend these observations in 3 parts. In Part I, we address the role of subjective understanding in determining attitudinal strength. This we define as the modular (i.e., absolute) extent to which an individual's stated attitude deviates from neutral, a measure that is likely to be reflective of related components, such as attitudinal longevity or influence on behaviour [12]. While in the above studies [4–6], attitudinal position is predicted by an interaction between objective and subjective knowledge, there is a more general underlying possibility, namely that attitudinal strength (either positive or negative) is predicted by subjective understanding. This model proposes that "the more I believe I understand the science, the more confident I will be in my acceptance or rejection of it." Put differently, psychological self-confidence to hold an extreme attitude (either positive or negative) implies strong subjective belief in the correctness of one's own understanding [8,9]. This would also accord with the notion that factual (objective) and subjective knowledge are separate constructs [13,14] with only a moderate correlation between them [15–17]. If this is true, it provides the basis for a potentially general model of strength of attitude that envisages this to be dependent on the extent to which individuals think they understand the science, rather than how much they do understand the science.

The prior study of a similar structure to ours (see also [6]) considered "no opposition" as one extreme of attitudinal distribution [4] and was thus unable to address this broader question concerning those with strongly positive attitudes (i.e., there was no differentiation between those who were strongly supportive, weakly supportive, or neutral). Similarly, the largest scale recent consideration focused on the trends for those whose attitudes were not aligned with the scientific consensus [6]. In focusing on those that reject accepted science, these studies leave open what we might think of as the natural history of those who also strongly accept the science. They did however report that, in some instances, as degree of opposition declines, subjective understanding may modestly increase [4,6]. In Part I, we employ our recent probability survey data from over 2,000 randomly selected United Kingdom adults to address the relationship between attitudinal strength and subjective understanding. We asked to what extent the respondents agree/disagree with the following statements:

1. Many claims about the benefits of modern genetic science are greatly exaggerated.

2. Those in charge of new developments in genetic science cannot be trusted to act in society's interests.

We refer to question 1 as "Hype," while we regard question 2 as being about "Trust." Both were measured on a 5-point scale (strongly agree, agree, neither agree or disagree, disagree, and strongly disagree). Responses we scored −2 to +2 with +2 implying a more accepting position (i.e., strongly disagree regarding both questions). Importantly, with this scoring system, we symmetrically recover attitudinal position, rather than focusing on degrees of negativity. Responses to the 2 questions are moderately correlated (rho = 0.44, $P < 10^{-15}$) suggesting that they are detecting more general attitudinal positions. Attitude strength is defined as the modulus (i.e., absolute value) of these values (scale 0 to 2). Subjective scientific understanding was assessed using 6 questions (see Methods).

The model predicts that attitude strength should be positively correlated with subjective understanding and that this will be robust to covariate control (Prediction 1a). We consider age, religiosity, political identity, and educational attainment as covariates. Both educational attainment [1] and religiosity [18] are known to relate to science attitude. For more polarised sciences, but not typically for genetical issues such as GM, political identity can predict

attitudinal position [1]. As age may well covary with both, and is likely to be relevant in the context of COVID, where the old were especially affected, we also include it as a covariate. Such a test need not establish that both extremes of attitudinal position score highly for subjective understanding. Thus, a second prediction of this model is that the same trends are seen as the distribution moves towards both extreme positive and extreme negative attitudes. We therefore ask whether the function relating attitudinal position, measured symmetrically, to subjective understanding is approximately U shaped. If so, the function should be better fitted by a quadratic model than by a linear model (Prediction 1b). However, an improved quadratic fit may yet be obtained if only one of the 2 attitudinal extremes is associated with increased subjective understanding, the other half of the data being uncorrelated. Assuming the quadratic fit is better, we additionally ask whether those with an attitudinal score greater than or equal to zero (neutral) show a positive correlation between subjective understanding and attitude that reverses to a negative correlation for those with an attitudinal score less than or equal to zero (Prediction 1c).

In Part II, we address the direction (valance) of attitudinal position. The prior studies found that those with subjective overconfidence (that is, low objective knowledge but high subjective understanding) tend to have more negative attitudes towards GM technologies [4] and vaccines [5]. This interaction is not logically necessary: it could be that strongly negative and positive attitudinal positions are associated with higher knowledge levels, as seen in climate change debates [1,2]. Alternatively, highly knowledgeable experts, who know they are experts, may also take negative positions (e.g., geneticists who are critical of human germline editing) [19]. Given our questions 1 and 2 above, which make no mention of specific technologies, we ask (Part IIa) whether we observe the same effects. This enables us to ascertain whether the prior recent results [4,5] reflect OSD-mediated antagonism to specific technologies or are more general. As objections to specific technologies often focus on details specific to each technology, such as higher pesticide use associated with GM crops (see e.g., [20]) or misinformation about a relationship between vaccines and autism [5], this allows us to establish whether these technology-specific objections are core or conceal a deeper mistrust, at least for some.

To address this issue, we additionally consider 12 true/false "textbook" science questions (see also S1 Table). Individuals are then given a score from −1 to +1, zero reflecting responses no better than chance. The model predicts that a quadratic fit of subjective to objective scores should fit better than a linear fit, reflecting a pool with low factual knowledge but high subjective understanding (Prediction 2a). Further, and crucially, it predicts that those individuals with low knowledge, but high subjective scores, should be disproportionately in classes with negative attitudes (Prediction 2b). Thus, the difference between objective knowledge and subjective understanding should be predicted by attitudinal position. Given our scoring system (Methods), we define this variable as:

OSD = knowledge score—self-assessment score.

The direction of this effect, if in accord with prior results [4,5], is expected to be such that as attitude becomes more negative the deficit becomes larger. With our metric, we predict that OSD should be positively correlated with non-modular attitudinal position (Prediction 2b). We ask whether this is robust to covariate control (Prediction 2c) and in the process determine independent predictors of this deficit (Prediction 2c corollary). As prior comparable analysis considered attitudinal positions with a maximum of no objection [4], we also seek to replicate their method by comparing the slopes of the subjective understanding and the objective knowledge scores as a function of attitude for cases where attitude score is less than or equal to zero (Prediction 2d). We expect the most negative attitudes among those who are low in factual knowledge but high in subjective understanding, hence a difference in slope.

In Part IIb, we seek to broadly repeat the result that OSD predicts attitudinal position as regards GM (replication test 1) and vaccines (replication test 2) [4–6]. The structure of the GM question also enables us to address the attitude strength issue. These tests are important in 2 regards. First, to see whether the overconfidence effect [4–6] generalises to technology nonspecific questions, it is also important to see that within our data, we see the technology-specific effect. Were this not the case then the relationship with prior claims would be uncertain. Second, the prior claims are relatively new and their resilience to replicational studies uncertain, although recent work supports strong generality across multiple scientific issues, but not all [6]. In seeking to reproduce prior results, we refer to these as replications, but they are far from precise technical replications. Indeed, in previous studies concerning vaccination, the issue at concern was knowledge about autism and its relationship to attitudes towards MMR vaccination [5]. Here, we ask about attitudes to COVID vaccines for which autism has not been a central concern. We also measure OSD rather than subjective overconfidence, defined by asking whether people considered themselves as or more expert than doctors [5]. Our tests are nonetheless helpful in seeking to generalise the prior results.

We find evidence for high OSD predicting negative attitudes for all 4 questions. Remaining agnostic as to the cause, we ask (Part III) whether predictors of OSD are different for the 4 different questions. Were they different, then the quest for a universal mechanistic narrative relating OSD to attitudinal valence may be complex (if not hopeless) and instead contingent on the question, even when these are close relatives.

In our data, the higher the attitudinal positivity, the higher the scientific knowledge (S1 Results) and the higher the maximum educational attainment (S1 Results). We see no evidence that the population is divided into 2 discrete camps of opposing attitudinal positions (S1 Results), nor for political positions to be highly divergent between those extremely supportive and those extremely negative (S1 Results). Indeed, as regards the Trust question both extremes tend to be more left wing than more moderate positions (S1 Results), although the same trend is not seen for the Hype question. These features suggest the responses are not like those to highly polarised/politicised science [1] and better resemble the classical low knowledge /negative attitude trend [3].

We report 2 novel results. First, we observe that individuals with more extreme attitudes, in either positive or negative directions, tend to be more confident that they understand genetic science. This provides a potentially general underpinning model for attitudinal strength the generality of which is yet to be determined. Second, while replicating the finding that those reporting negative attitudes are mistaken in their belief that they understand the science, we find this to be observed both with and without reference to specific genetic technologies. Objection to specific technologies may thus for some be a smokescreen concealing an underlying negativity underpinned by unjustified self-confidence. We discuss implications of these results for effective science communication and more broadly.

## Results

### Part I: Attitudinal strength is predicted by subjective understanding

**Prediction 1a. Attitude strength correlates with subjective understanding controlling for covariates.**   Attitude strength is positively correlated with subjective understanding (Spearman rank test: Trust: rho = 0.24, $P = 2 \times 10^{-16}$; Hype, rho = 0.30, $P = 2 \times 10^{-16}$). This is robust to covariate control (Table 1: Trust, partial rho = 0.226, $P = 4 \times 10^{-25}$; Hype partial rho = 0.27, $P = 5 \times 10^{-37}$). The attitude strength scores are otherwise only predicted by educational attainment with higher attainment associated with stronger attitudes (Table 1). We conclude that stronger attitudes are associated with stronger subjective assessment of understanding.

**Table 1. Correlation and partial correlation analysis of between modular Trust score, modular Hype score, and SAU controlling for age, religiosity, political identity, and educational attainment.**

| Trust score | | | | | | |
| --- | --- | --- | --- | --- | --- | --- |
| | Education | Age | Religiosity | Politics | SAU | \|Trust\| |
| Education | - | **−0.14** | 0.04 | **−0.26** | **0.28** | **0.10** |
| Age | **−0.08** | - | **0.20** | **0.24** | **−0.13** | −0.01 |
| Religiosity | **0.10** | **0.17** | - | **0.17** | −0.03 | 0.00 |
| Politics | **−0.21** | **0.18** | **0.15** | - | **−0.17** | **−0.08** |
| SAU | **0.23** | **−0.08** | −0.01 | −0.08 | - | **0.24** |
| \|Trust\| | 0.02 | 0.03 | 0.01 | **−0.05** | **0.22** | - |
| **Hype score** | | | | | | |
| | Education | Age | Religiosity | Politics | SAU | \|Hype\| |
| Education | - | **−0.14** | 0.04 | **−0.26** | **0.28** | **0.15** |
| Age | **−0.08** | - | **0.20** | **0.24** | **−0.13** | −0.01 |
| Religiosity | **0.10** | **0.17** | - | **0.17** | −0.03 | 0.00 |
| Politics | **−0.21** | **0.18** | **0.15** | - | **−0.17** | **−0.13** |
| SAU | **0.22** | **−0.08** | −0.01 | **−0.06** | - | **0.30** |
| \|Hype\| | **0.05** | **0.05** | 0.01 | **−0.09** | **0.27** | - |

The values above the diagonal are the spearman rho values for each comparison. Those below the diagonal are the same pairwise partial correlations controlling for all other variables. All values in bold are significant (at under 0.05). All nonsignificants are $P > 0.05$; $N = 2,051$. All scripts and data are available at doi: 10.5281/zenodo. 7289133.

SAU, subjective science understanding.

**Predictions 1b/1c: Extreme negative and positive attitudes towards genetics are associated with subjective understanding.** The model requires that attitude strength increases with subjective understanding for both those with negative and positive attitudes. In accord with this, we observe a U-shaped distribution between subjective understanding and both Trust and Hype (Fig 1). As predicted by the model (prediction 1b), a quadratic fit is highly significantly better than a linear fit in both instances: Trust: adjusted $R^2$ for quadratic model = 0.066; for linear model = 0.025; Hype: adjusted $R^2$ = 0.087 for quadratic model, for linear model = 0.035; $P < 10^{-16}$ in both cases for significance of difference between quadratic and linear fit.

Given that there is a relatively low number of individuals in the class considered most negative ($N = 47$ for Trust, $N = 29$ for Hype), it is possible that the U-shaped function is an artefact of skewed sampling with larger classes disproportionately influencing the form of the curve. To address this, we resample the data, this time allowing equal numbers in each classification (−2, −1, 0, etc.), this number being the size of the smallest of the classifications (in these 2 incidences, this is the most negative class). Thus, if there are 47 in the least trusting class, we resample without replacement 47 from each of the other 4 classes generating a balanced sample $47 \times 5 = 235$ long. We then repeat the test for linear and quadratic fits and ask in what proportion of resamplings is a quadratic fit significantly better than the linear fit at $P < 0.05$. Repeating for 10,000 randomisations, we find for Trust that 100% of randomisations have an improved quadratic fit, and for Hype, 99.14% have an improved quadratic fit. We conclude that the observed improved quadratic fit is not an artefact of skewed samples sizes.

Given this improvement of the quadratic fit, we additionally consider the correlation between subjective understanding and attitude score for both halves of the distribution separately (Prediction 1c). This confirms the trends (Trust $> = 0$, rho = 0.27, P = $2 \times 10^{-16}$; Trust $< = 0$, rho = −0.13, $P = 8.7 \times 10^{-6}$; Hype $> = 0$, rho = 0.33, $P = 2 \times 10^{-16}$; Hype $< = 0$,

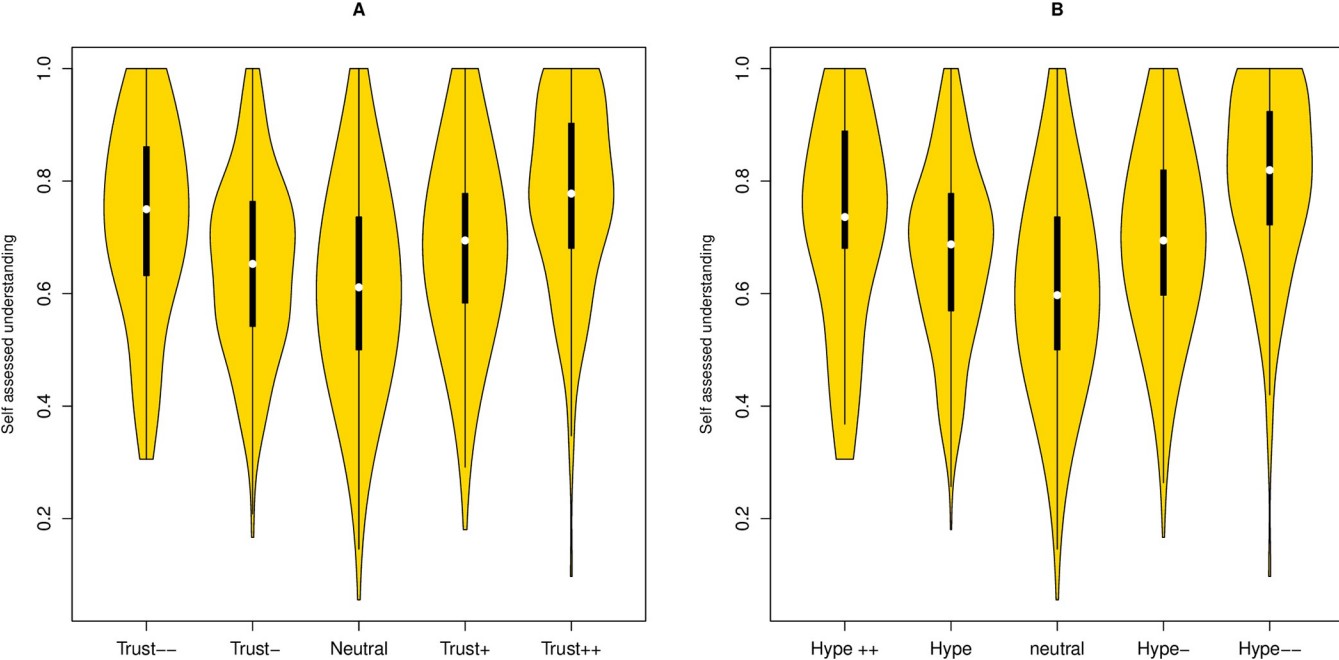

**Fig 1. Subjective science understanding as a function of attitudinal position for (A) Trust, (B) Hype.** Both are plotted with more rejectionist views to the left (Trust—, Hype ++). Scoring the x values, −2 to +2, a quadratic fit is highly significant in both instances ($P < 10^{-16}$ in both cases). All scripts and data are available at doi: 10.5281/zenodo.7289133.

rho = −0.17, $P = 9 \times 10^{-10}$). We conclude that as subjective understanding increases so too does attitudinal extremity, in both positive and negative directions.

## Part II: Objective–subjective deficit as a predictor of negative attitudes

**Part IIa.** *Prediction 2a*: *An excess with low knowledge but high subjective understanding.* The above establishes a general attitude strength-subjective understanding effect but establishes nothing about valence (i.e., directionality of attitude). If there exists a subpool of individuals with high subjective understanding but low knowledge, we expect a quadratic fit between the 2 parameters. Overall, we find a moderate correlation between subjective and objective measures (Fig 2: rho = 0.47, $P < 2 \times 10^{-16}$). This implies that, to a first approximation, the UK public have an accurate perception of their own scientific abilities. The quadratic fit is, however, a significantly better fit than a linear fit (linear model: Adjusted $R^2$ = 0.214; quadratic model: Adjusted $R^2$ = 0.221, the quadratic being the significantly better fit, $P = 8.7 \times 10^{-6}$). This is consistent with a subpopulation at low levels of scientific knowledge with higher subjective understanding scores than expected under a simple linear model, potentially compatible with a relatively rare low knowledge but high self-confidence subpopulation.

*Prediction 2b*: *The subjective-objective deficit is predicted by negativity of attitude, higher religiosity, and lower educational attainment.* For both Trust and Hype, the higher the acceptance, the higher the scientific knowledge (Spearman rank tests: Trust, rho = 0.21, $P = 2 \times 10^{-16}$; Hype rho = 0.29, $P < 2 \times 10^{-16}$) (S1 Fig). Given this and the U-shaped function in prediction 1b/1c, we might expect an OSD that covaries with attitudinal position. A negative score implies overconfidence, i.e., a deficit in scientific knowledge below subjective levels. As previously reported [21–23], most people are overconfident having a negative OSD (see Fig 2).

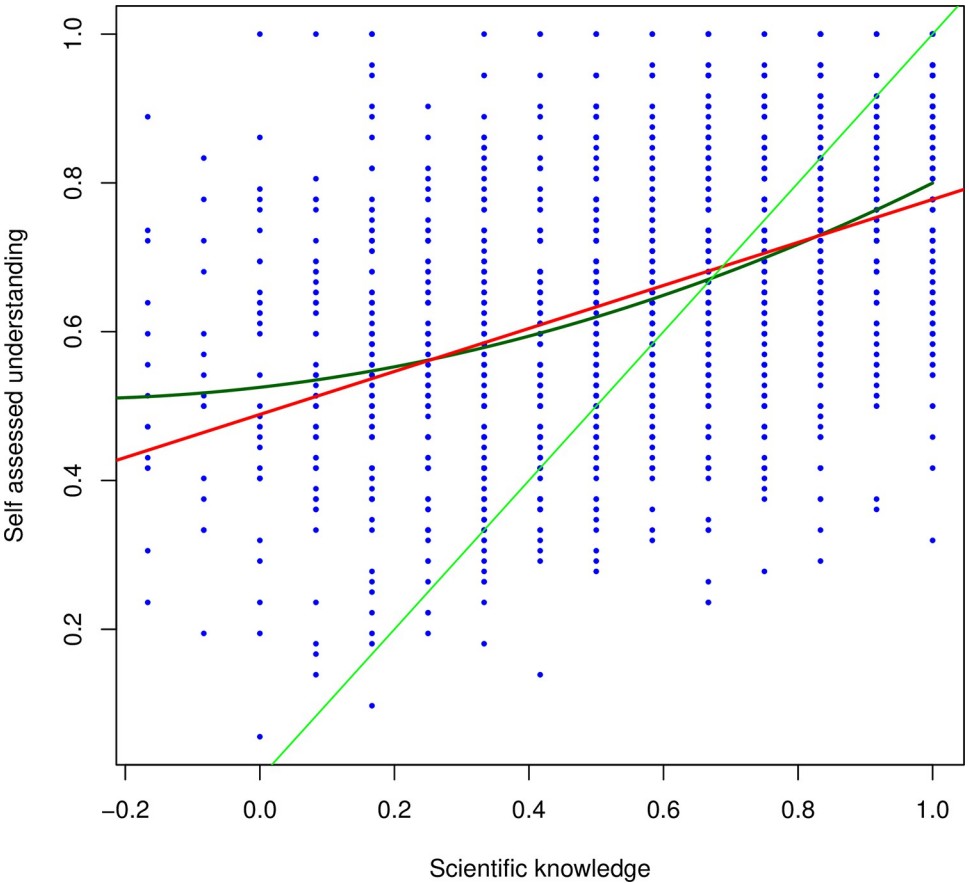

**Fig 2. Subjective science understanding and objective science knowledge covary.** The linear fit is shown in red, quadratic in dark green. The light green line is the line of slope 1 to define OSD. All scripts and data are available at doi: 10.5281/zenodo.7289133. OSD, objective–subjective deficit.

Given prior trends (Figs 1 and S1), OSD should become more negative as attitudinal position becomes more negative. This is indeed observed (Fig 3: Trust, rho = 0.10, $P = 3.0 \times 10^{-6}$; Hype: rho = 0.17, $p < 5.9 \times 10^{-15}$). Heterogeneity with a low large deficit for the strong rejectors is also seen using metric of population relative OSD based on differences in Z scores [4] (ANOVA: Trust, $P = 1.6 \times 10^{-5}$; Hype, $P = 2 \times 10^{-8}$). We conclude that individuals with a greater deficit (i.e., more negative OSD) are more likely to hold negative attitudes towards genetics.

*Prediction 2c*: *OSD-attitude correlations are robust to covariate control*. The OSD-attitude correlations may be explained as a consequence of covariates. We again consider 4 parameters: age, religiosity, political identity, and educational level. The OSD-attitude result is robust to covariate control (Trust v OSD, partial rho = 0.08, $P = 1.6 \times 10^{-4}$; OSD Hype, partial rho = 0.14, $P = 3 \times 10^{-10}$ (Table 2)). We conclude that more negative attitudes are associated with low objective knowledge compared to subjective knowledge and that this trend is not explained by the covariates age, religiosity, political identity, and educational level.

Independent of the attitudinal positions, we find that a knowledge deficit (overconfidence) is predicted by higher religiosity (rho = −0.09), lower educational attainment (rho = 0.15), more right-wing attitudes (rho = −0.12), and lower age (rho = 0.06) (Prediction test 2 corollary: Table 3). These results are robust to the relative OSD metric [4] (Table 3), excepting for the OSD education correlation.

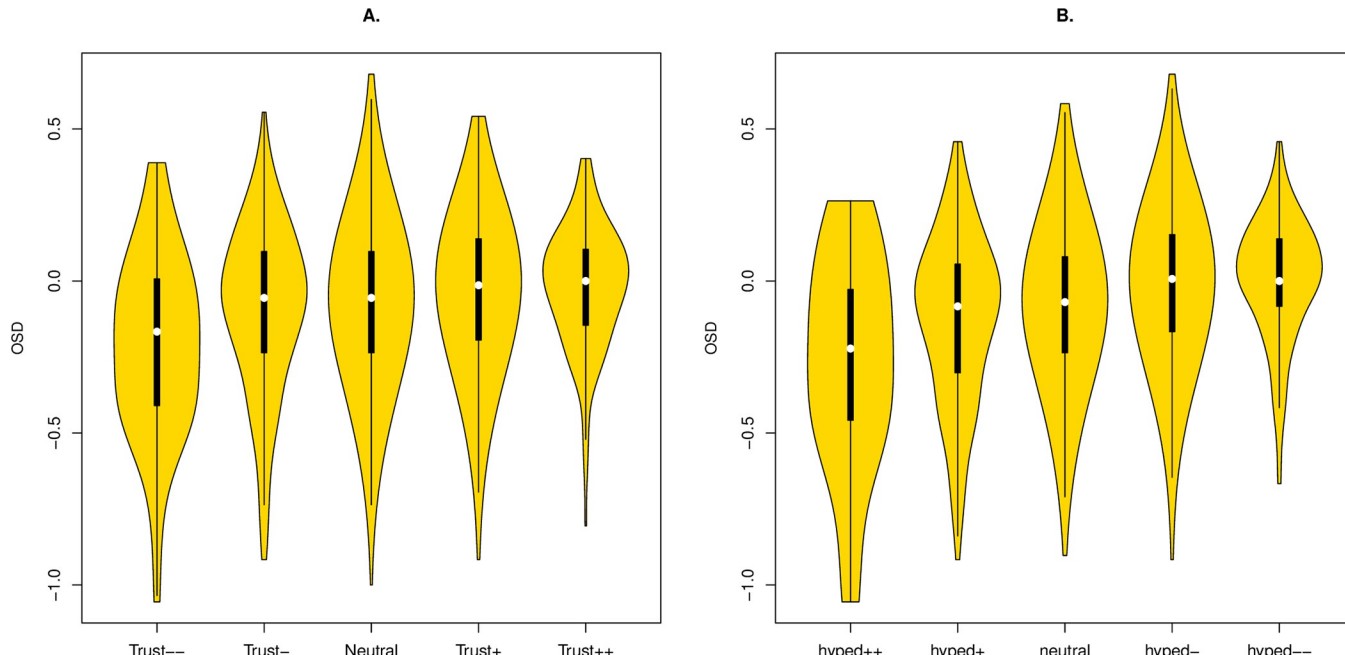

**Fig 3. OSD as a function of attitude for (A) Trust and (B) Hype.** Both are plotted with more rejectionist views to the left (Trust—, Hype ++). All scripts and data are available at doi: 10.5281/zenodo.7289133. OSD, objective–subjective deficit.

Considering those cases where the attitudinal score is less than or equal to zero, we find that the slope of the subjective understanding versus attitude score is significantly different for Trust and Hype, (and more negative) than for the objective knowledge (Prediction 2d;

**Table 2. Correlation and partial correlation analysis of age, religiosity, educational attainment, political identity, gap, and Trust and Hype scores.**

| Trust score | | | | | | |
|---|---|---|---|---|---|---|
| | Education | Age | Religiosity | Politics | OSD | Trust |
| Education | - | **−0.14** | 0.04 | **−0.26** | **0.15** | **0.11** |
| Age | **−0.11** | - | **0.20** | **0.24** | **0.06** | −0.03 |
| Religiosity | **0.11** | **0.18** | - | **0.17** | **−0.09** | **−0.05** |
| Politics | **−0.23** | **0.19** | **0.14** | | **−0.12** | **−0.05** |
| OSD | **0.13** | **0.12** | **−0.09** | **−0.09** | | **0.10** |
| Trust | **0.09** | −0.01 | −0.04 | −0.01 | **0.08** | - |
| **Hype score** | | | | | | |
| | Education | Age | Religiosity | Politics | OSD | Hype |
| Education | - | **−0.14** | 0.04 | **−0.26** | **0.15** | **0.15** |
| Age | **−0.11** | - | **0.20** | **0.24** | **0.06** | −0.04 |
| Religiosity | **0.12** | **0.18** | - | **0.17** | **−0.09** | **−0.07** |
| Politics | **−0.22** | **0.19** | **0.13** | - | **−0.12** | **−0.15** |
| OSD | **0.12** | **0.12** | **−0.09** | **−0.07** | - | **0.17** |
| Hype | **0.11** | 0.0 | **−0.05** | **−0.09** | **0.14** | - |

The values above the diagonal are the spearman rho values for each comparison. Those below the diagonal are the same pairwise partial correlations controlling for all other variables. All values in bold are significant (at under 0.05). All nonsignificants are $P > 0.05$; $N = 2,051$. All scripts and data are available at doi: 10.5281/zenodo. 7289133.

OSD, objective–subjective deficit.

**Table 3. Correlation and partial correlation analysis of age, religiosity, educational attainment, and OSD score.**

| Absolute gap metric | | | | | |
|---|---|---|---|---|---|
| | Education | Age | Religiosity | Politics | OSD |
| Education | - | **−0.14** | 0.04 | **−0.26** | **0.15** |
| Age | **−0.11** | - | **0.20** | **0.24** | **0.06** |
| Religiosity | **0.11** | **0.18** | - | **0.17** | **−0.09** |
| Politics | **−0.23** | **0.19** | **0.14** | - | **−0.12** |
| OSD | **0.14** | **0.12** | **−0.10** | **−0.09** | - |
| Z-based metric | | | | | |
| | Education | Age | Religiosity | Politics | OSD |
| Education | - | **−0.14** | 0.04 | **−0.26** | 0.02 |
| Age | **−0.10** | - | **0.20** | **0.24** | **0.11** |
| Religiosity | **0.10** | **0.18** | - | **0.17** | **−0.06** |
| Politics | **−0.24** | **0.19** | **0.15** | - | −0.03 |
| OSD | 0.034 | **0.13** | **−0.08** | −0.04 | - |

The values above the diagonal are the spearman rho values for each comparison. Those below the diagonal are the same pairwise correlations controlling for all other variables. All values in bold are significant (at under 0.005). All nonsignificants are $P > 0.05$; $N = 2,051$. All scripts and data are available at doi: 10.5281/zenodo.7289133. OSD, objective–subjective deficit.

Table 4). This accords with the hypothesis that as attitudes become more negative, subjective understanding increases more than does objective knowledge.

**Part IIb.** Existing evidence suggests that high OSD predicts negative attitudes for both the GM and the vaccine issues [4,5]. Here, we ask whether our data replicates both OSD effects. In the case of GM, we also investigate attitudinal strength (Prediction group 1) as the structure of the test is comparable to that employed above.

**Replication 1, GM attitudes.** To address the GM issue, after asking questions 1 (Hype) and 2 (Trust), we asked: "On balance, the advantages of genetically modified (GM) foods outweigh any dangers." This was scored −2 to +2, with, as before, more negative attitudes being more negative in score. The distribution is approximately normal (S1A Fig). The attitude strength predictions are all upheld: (1a) attitude strength is positively correlated with subjective understanding (rho = 0.20, $P < 2 \times 10^{-16}$), (1b) a U-shaped function relates subjective understanding and attitudinal position (Fig 4A: quadratic fit, adjusted $R^2 = 0.057$; $P = 2 \times 10^{-16}$; linear fit adjusted $R^2 = 0.15$, $P = 9.1 \times 10^{-9}$, P for improved fit = $2.2 \times 10^{-16}$), and (1c) subjective understanding is correlated with attitude in the expected manner (GM > 0, rho = 0.27, $P = 2 \times 10^{-16}$; GM< = 0, rho = −0.116, $P = 7 \times 10^{-6}$). In none of 10,000 resamplings, in which all classes are the length of the smallest class (strongly accepting in this case), is the improved quadratic not found.

The OSD/valence predications are also largely upheld. The modular attitude strength verses subjective understanding (rho = 0.20, $P < 2 \times 10^{-16}$) is a stronger effect that the relationship between objective knowledge and attitudinal position (Spearman rank correlation rho = 0.14,

**Table 4. Test of difference in slope for the subjective and objective knowledge scores as a function of attitude for cases where attitudes score is = < 0.**

| Attitude | Slope subjective | SEM | Slope objective | SEM | t | P | df |
|---|---|---|---|---|---|---|---|
| Trust | −0.0444 | 0.0090 | −0.0026 | 0.0150 | −2.3866 | 0.0085 | 2,250 |
| Hype | −0.0609 | 0.0099 | 0.0009 | 0.0163 | −3.2462 | 0.0006 | 2,458 |

All scripts and data are available at doi: 10.5281/zenodo.7289133.

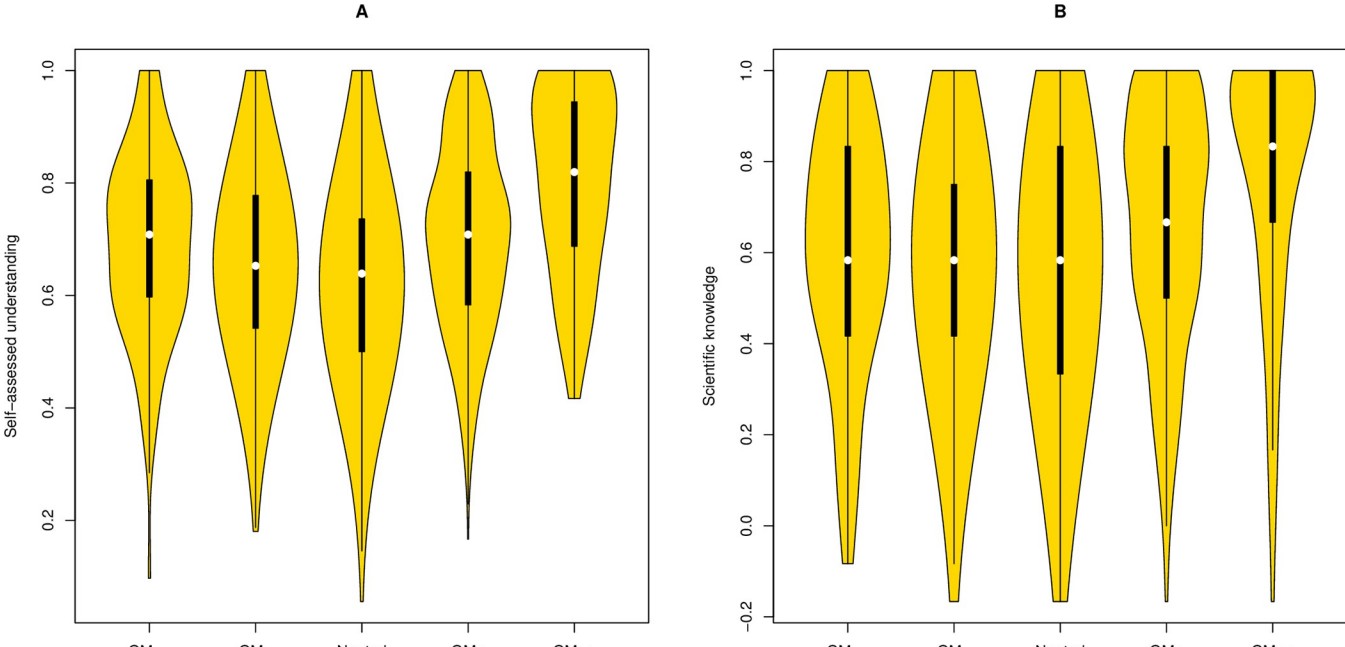

**Fig 4. The relationship between position on GM and (A) subjective understanding and (B) objective knowledge.** All scripts and data are available at doi: 10.5281/zenodo.7289133. GM, genetic modification.

$P = 4.5 \times 10^{-10}$ (Fig 4B) and consequently, OSD is predicted by attitudinal position (rho = 0.08, $P = 2.9 \times 10^{-4}$), this being robust to covariate control (partial rho = 0.081, $P = 2.4 \times 10^{-4}$) (Table 5). For frequencies see S1A Fig, for educational attainment see S1B Fig, for political identify see S1C Fig. However, the one exception is that when we restrict analysis to only instances where the attitude score is less than or equal to zero, the slopes of the 2 lines are not significantly different, although the trend is in the expected direction ($P = 0.16$, $t = -1.014$, df = 2,982; slope subjective = −0.0321 +/− 0.0070; slope for objective = −0.0186 +/− 0.0114) (but see also Discussion).

These results as regards OSD/valence thus largely replicate a prior study [4], excepting that the more extreme rejectors of GM technology are not the least knowledgeable; they are simply less knowledgeable than the extreme acceptors. We return to this issue in Part III.

**Replication 2, vaccine attitudes.** Given that this survey was performed during the pandemic, with the vaccine roll out having commenced in the UK 5 months prior, we chose to

**Table 5. Correlation and partial correlation analysis of age, religiosity, educational attainment, and GM score.**

|  | Education | Age | Religiosity | Politics | OSD | GM |
|---|---|---|---|---|---|---|
| Education | - | **−0.14** | 0.04 | **−0.26** | **0.15** | **0.05** |
| Age | **−0.11** | - | **0.20** | **0.24** | **0.06** | **−0.11** |
| Religiosity | **0.11** | **0.18** | - | **0.17** | **−0.09** | **−0.05** |
| Politics | **−0.23** | **0.19** | **0.14** | - | **−0.12** | −0.03 |
| OSD | **0.14** | **0.13** | **−0.10** | **−0.09** | - | **0.08** |
| GM | 0.02 | **−0.10** | −0.02 | 0.02 | **0.080** | - |

The values above the diagonal are the spearman rho values for each comparison. Those below the diagonal are the same pairwise partial correlations controlling for all other variables. All values in bold are significant (at under 0.05); $N = 2,051$. All scripts and data are available at doi: 10.5281/zenodo.7289133.

GM, genetic modification; OSD, objective–subjective deficit.

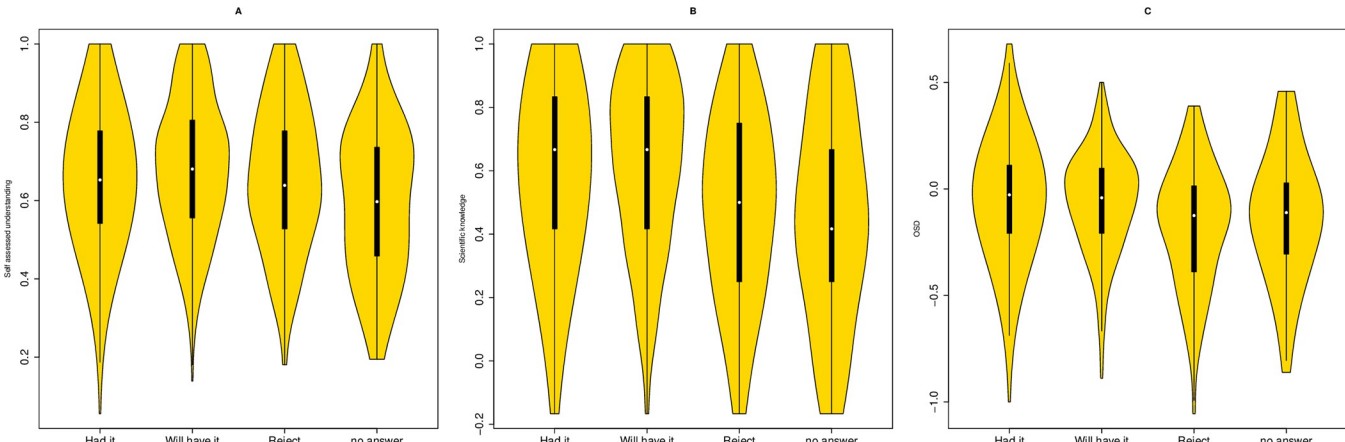

**Fig 5. Vaccine stance and (A) subjective understanding, (B) scientific knowledge, and (C) OSD.** For statistics, see S2 Table. All scripts and data are available at doi: 10.5281/zenodo.7289133. OSD, objective–subjective deficit.

examine behaviour rather than asking about attitudinal position. We asked correspondents whether they had taken the vaccine, intend to but have not yet, have no intention to, or prefer not to say. The structure of this question is such that we cannot evaluate attitude strength issues.

We observe heterogeneity between the 4 groups in their subjective understanding (Fig 5A, ANOVA $P = 0.0073$). However, post hoc Tukey tests reveal differences between those preferring not to answer and those who will be but are not yet vaccinated exclusively ($P = 0.006$). Those rejecting the vaccine are no different to those who have taken it ($P = 0.99$) and those who will ($P = 0.60$). By contrast, with respect to knowledge, there is again heterogeneity (Fig 5B, ANOVA, $P = 3.6 \times 10^{-6}$) with significant differences between those not getting vaccinated and both groups that will (Tukey post hoc $P < 0.001$ in both cases). Indeed, the only 2 comparisons that are not significant are between those who will not and those who prefer not to disclose ($P = 0.96$) and the 2 groups that will or have had it ($P = 0.82$). As then expected, given these results, those rejecting the vaccine have a larger OSD than either group of vaccine acceptors (Fig 5C. ANOVA: $P = 7.7 \times 10^{-5}$: Tukey post hoc $P < 0.002$ for both rejector v acceptor comparisons—note no other comparisons are significant). For full Tukey statistics, see S2 Table. We conclude that the OSD predicts vaccine acceptance, this replicating prior analysis [5,6].

COVID vaccine hesitancy is centred on the younger age groups (mean age of those who have or will take the vaccine if offered = 49, of those declining = 38, Mann–Whitney U test, $P = 6 \times 10^{-9}$). There is no difference between those who will or have taken the vaccine and those who say they will not in either educational attainment (Mann–Whitney U test, $P = 0.18$) or in religiosity (Mann–Whitney U test, $P = 0.34$) although those refusing the vaccine have a relatively high proportion without educational qualifications. In political identity, there is heterogeneity between classes (ANOVA, $P = 2.5 \times 10^{-8}$) but only 2 pairwise comparisons are significant: those who have had the vaccine are more conservative than those who will have (Tukey test, $P < 0.00001$) and those who have yet to be vaccinated but will be are more left wing than those who will not have the vaccine. For frequencies see S2A Fig, for sex ratios see S2B Fig (there is no heterogeneity chi sq = 5.4, $P = 0.14$, df = 3), for educational attainment see S2C Fig, for political identify see S2D Fig.

## Part III: Heterogeneity of demographic predictors of OSD

For all 4 questions, we report that stronger negativity of attitude is associated with a larger OSD. Were there a simple underlying cause, one might expect that all 4 might show similar

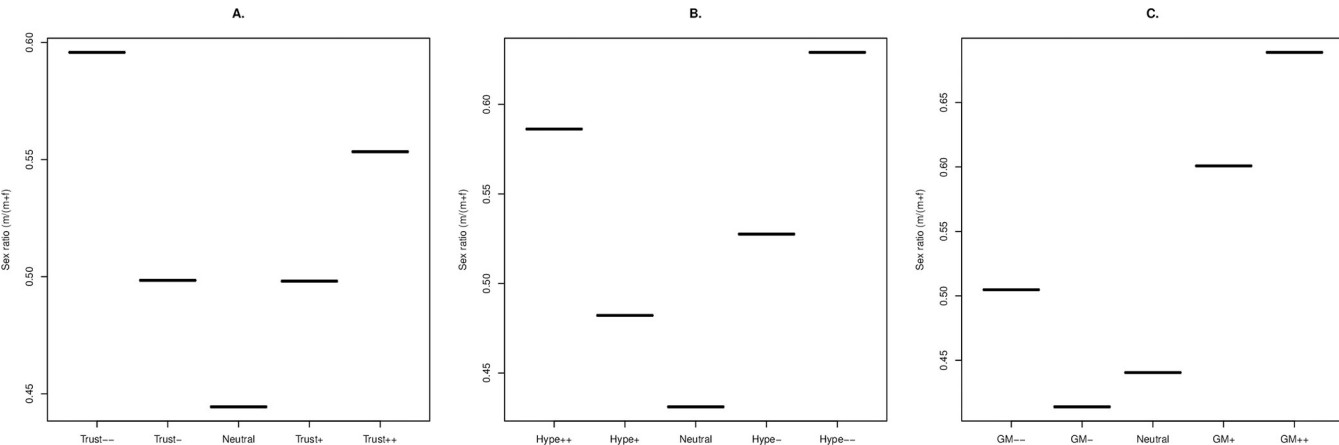

**Fig 6. Sex ratio as a functional of attitudinal position for (A) Trust, (B) Hype, and (C) GM benefits.** All are plotted with more negative views to the left. In both instances, there is significant heterogeneity in the sex ratio between the attitudinal groups (Trust: chi squared = 10.9, df = 4, $P$ = 0.028; Hype: chi squared = 27.6, df = 4, $P = 1.5 \times 10^{-5}$; GM chi squared = 58.4 df = 4, $P = 6 \times 10^{-12}$). Men are more likely to adopt more extreme attitudinal positions for Trust and Hype but only extreme positive attitudes for the GM issue. All scripts and data are available at doi: 10.5281/zenodo.7289133. GM, genetic modification.

underlying demographic predictors. This appears not to be the case. As regards the COVID vaccine the strong predictor was age with younger subjects being more negative/hesitant. As the risk-benefit calculation in this incidence covaries with the age profile, this we suggest is an understandable outlier (no other question revealed a significant age effect in this direction).

Perhaps then, the more intriguing question is why the more abstract Hype and Trust questions—which have the same demographic predictors—seem to diverge from the GM question. That the GM questions might be exceptional is suggested by the fact the Hype and Trust responses correlate with each other much more robustly than either does to the GM question (Spearman rank correlation: Hype v Trust, rho = 0.44, GM v Trust 0.19, GM v Hype 0.18: for all, $P < 10^{-15}$). This suggests that asking specifically about GM is associated with a somewhat different response.

Several further differences between the GM attitude questions and the nonspecific Trust and Hype questions are notable. First, there are many more (approximately 5%) in the strong rejecting class than in the prior 2 strong rejector classes, these being 1% to 2% (S1A Fig). Second, this class has approximately 50:50 sex ratio, different to that for the other 2 (Fig 6). Third, for the GM question, education attainment is a much weaker predictor of attitudinal position and is indeed not significant on partial correlation analysis (Table 5, see also S1B Fig) nor on multitest correction. Further, unlike the prior 2 questions, age is a strong predictor (older people are more opposed to GM), before and after partial analysis (Table 5). Fourth, GM attitude is not correlated with political identity (as previously found [1]) while for both Trust and Hype, there is a tendency for those more oppositional to science to be more right wing (Table 2). Fourth, when we restrict analysis to cases where the attitudinal score is less than or equal to zero, we see no significant difference in the slopes relating knowledge score and subjective understanding to attitude for the GM issue. Fifth, many fewer of the 12 science knowledge questions are predictive of GM position than seen for the 2 more abstract questions and correlations are generally much weaker (mean rho for Trust = 0.094, Hype = 0.12, GM = 0.058: S1 Table).

A possible explanation for some of this difference is that among those most negative to GM is, in addition to those being negative as regards Trust and Hype, a subgroup of older well-educated individuals taken from all political persuasions. A coherent model then, is that there

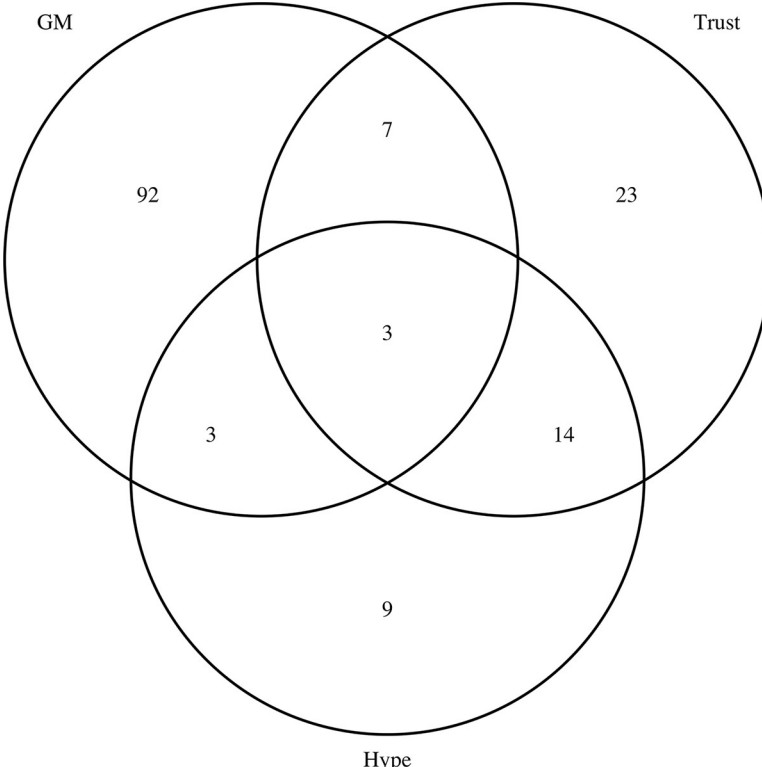

**Fig 7. Venn diagram of numbers in each group expressing strongly negative attitudes to 3 questions.** All scripts and data are available at doi: 10.5281/zenodo.7289133. GM, genetic modification.

exists a group of sceptics whose responses are not technology specific (revealed by the Hype and Trust questions) and a further demographically different group that are concerned by specific technologies. Consistent with this, employing the dip test [24,25], there is a significant tendency towards non-unimodality of knowledge scores in the GM strong rejectors ($P = 0.00109$), but not otherwise (S3 Table). This could in turn explain the educational profile (S1B Fig) and weak correlation with educational attainment that is not robust to covariate control as this larger pool is well educated. In addition, when we analyse which individuals report strong negativity in any of the 3 parameters (Trust, Hype, and GM), we find considerable overlap between the Trust and Hype groups and a large population (92 of 105) of the GM rejectors that do not report strong negativity as regards the Hype and Trust issues (Fig 7). The intersection between Hype and Trust is highly deviated from null ($Z > 20$), while that between GM attitude and the other 2 is less deviant from null ($Z$ in range approximately 4 to 6) (Table 6). While the size of the three-way intersection ($N = 3$) is modest in absolute terms it is highly significant ($Z > 16$, Table 6). The data thus support a model that the Hype and Trust questions are capturing a generalised antipathy towards science that is independent of specific technology, while negativity as regards the GM question recruits from both this pool of those generally anti-science but mostly those specifically anti-GM. For those showing both negativity as regards GM and the Hype/Trust questions, objections to GM based on technology alone may be considered a smokescreen concealing an underlying antipathy. Given these differences, we caution that the trends that we have observed, especially as regards covariate predictors, are not robustly generalisable. It also implies no universality to predictors of OSD as assayed via similar, but different, questions. It also suggests that there is a subgroup that object to GM but

**Table 6. Statistics for the two and three-way intersections as regards numbers expressing extreme negative views towards GM and as regards Hype and Trust issues (i.e., scoring −2 in any of these questions).**

| Comparison | N1 | N2 | N3 | Intersect | Mean_intersection_rand | SD_rand | Z | P |
|---|---|---|---|---|---|---|---|---|
| Hype v GM | 29 | 105 | NA | 6 | 1.4279 | 1.1564 | 3.9538 | 0.00225 |
| Hype v Trust | 29 | 47 | NA | 17 | 0.6501 | 0.7933 | 20.6112 | 0.0000 |
| Trust v GM | 47 | 105 | NA | 10 | 2.3001 | 1.4678 | 5.2461 | $1 \times 10^{-4}$ |
| Three way | 105 | 47 | 29 | 3 | 0.0316 | 0.1779 | 16.6843 | 0.0000 |

N1, N2, and N3 are the total numbers in each class. In the two-way interactions, the first named in the comparison column is N1. The intersection is the number of individuals in both groups. *P* was determined by randomisation. Here, from the source data 3 random sets were generated of sizes 29, 47, and 105, to match observed numbers in Hype, Trust, and GM, respectively. The size of relevant intersections (individuals shared/intersecting in the groups) was determined for each randomisation. This was repeated 100,000 times to generate, for each comparison, a distribution of the number of expected intersecting under a null model of no correspondence. From this vector, we determined how many had as great or greater than the observed intersection. *P* was then given as this number/100,000. To further assess magnitude of deviation from null, we determined the mean of the vectors of numbers intersecting under null and the SD of the same vector. From this, we could derive Z = (observed intersection–mean of randoms)/SD of randoms. All scripts and data are available at doi: 10.5281/zenodo.7289133.

GM, genetic modification.

also object when no technology is referenced for which any technological defence may well be a smokescreen. However, for most GM objectors the smokescreen hypothesis does not hold weight.

## Discussion

We asked 3 main questions in this study: whether attitude strength (both negative or positive) is associated with high subjective understanding (Part I), whether subjective understanding might additionally predict valence (as previously shown in [4–6,8]) by interaction with absolute knowledge (Part II), and whether any such trends are seen with and without reference to particular genetic technologies.

We repeatedly find that as one tends to higher degrees of subjective understanding, so attitudinal positions become more extreme. This supports the hypothesis that people with more extreme attitudes are more confident that they understand the science. In all analyses, this effect (measured by the correlation between modular attitude score and subjective score) is stronger than the much-studied attitude–knowledge score [3] and is robust to covariate control. The psychological model that sees a strong attitudinal position as being underpinned by a strong subjective belief in the correctness of ones' understanding is thus robustly supported.

One might question, however, why the trend that we see has not been reported previously. Recent comparable analyses, those of Light and colleagues [6] and Fernbach and colleagues [4], consider a scale in which one extreme is "no opposition," while our metric considers both extreme acceptance and extreme rejection. As a consequence, their metric is ill-suited to identification of strong self-confidence of those most accepting. Nonetheless, the results of both Light and colleagues and Fernbach and colleagues [4] are supportive of a tendency for subjective knowledge to increase as degree of opposition reduces. For example, scrutiny of the supplementary figures of Light and colleagues [6] suggests that at lower degrees of opposition, at least for some issues, there may be raised subjective understanding, what they refer to as a J-shaped curve: this is seen in their supplementary Figures 2 and 5, but not in their supplementary Figure 8. In Fernbach and colleagues [4], a similar J-shaped figure is seen (and supported by improved quadratic fit), indicative of increasing self-confidence of understanding in those with lower opposition. In contrast to these results, in our data, the strength of association between subjective understanding and attitude is stronger for those with a positive attitude

than those with a negative attitude. We conclude that the effect that we have identified was discernible in prior data but possibly under-appreciated owing to a metric that focused on those least accepting (and a tendency to perform and report linear analysis).

Given the positive correlation between attitude and factual knowledge, this implies an OSD among those most negative in attitude (Part II). As with the attitude strength/subjective understanding correlation, the OSD–attitude strength effect is also robust to covariate control. We additionally (mostly) replicate prior studies on OSD and negative attitudes to GM technologies [4] and vaccines [5] that report that strong opponents tend to be overconfident. Our results extend these prior results to suggest that we do not need to specify particular issues/technologies to observe the same effects. Indeed, the effects are more pronounced when specific technology is not mentioned. The specific exemplars then may instead be considered as special cases of more generalised antagonism towards science or scientific expertise. Objections centred on specificities of the technology may then be, at least for some, a smokescreen, hiding a more general antipathy. However, comparison with GM attitude suggests that for most GM objectors this is not a robust hypothesis.

### Why might strongly negative attitudes as regards Trust and Hype be associated with high OSD?

While we have argued that the demographics of OSD for the 4 questions are not congruent, for Hype and Trust they are similar. Those most negative for these 2 questions appear to be a small kernel (1% to 2%) of fundamentalist sceptics who are disproportionally male, less well educated, more right wing, and more religious. Why might low factual knowledge be especially associated with high subjective knowledge and then with a strongly negative attitude for this group?

In previous studies, authors speculate [4,5] about Dunning–Kruger type effects [10], whereby low knowledge may be associated with low analytical abilities that prevent understanding that one's knowledge is limited. We did not measure variables relevant to ascertainment of this specific hypothesis. Nonetheless, further dissection of our 12 factual questions suggests that such a model is insufficient. For both Hype and Trust, the 3 questions of the 12 true/false "textbook" questions most strongly correlating with these 2 attitudinal positions (S1 Table) are questions 4 ("By eating a genetically modified fruit, a person's genes could also become modified"), 5 ("All radioactivity is human-made"), and 8 ("Tomatoes do not naturally contain genes—genes are only found in genetically modified tomatoes"). In addition, answers to these 3 questions were strongly correlated (S4 Table). These are the only 3 questions that unambiguously address human interventions (or lack thereof). This suggests that the effect might be construed as more conspiratorial in nature. A conspiratorial aspect is also attractive in so much as it goes some way to explain the negativity [5]. This need not imply a lack of logical inference skills. Indeed, that those who consider that only GM tomatoes have genes also consider that it possible to take up genes from GM crops is consistent with a model of logical inference from incorrect understandings.

The possibility of conspiratorial thinking is broadly consistent with other features of this subpopulation. Indeed, the fact that they provided strongly negative attitudes when specific technologies were not mentioned, suggests a broader scepticism than seen for the GM or vaccine issues. Our data provides some further support for this broader scepticism. We presented the respondents with 15 diverse information/media outlets and asked which if any they trusted. Those reporting Trust in none of the outlets also tend to be those reporting little Trust in science (Mann–Whitney U test $P = 6 \times 10^{-14}$; mean science Trust score (range −2 to +2) of those trusting one of more media outlets = 0.38 +/−0.88 SD, $N = 1,814$ of those not trusting any = −0.084 +/−0.9, $N = 250$). This suggests that lack of trust in science is associated with a

more generalised lack of Trust in media sources more generally. Our results then also accord with the notion that individuals with low levels of both generalised interpersonal trust and of domain-specific knowledge may be especially likely to endorse conspiracy theories [26]. The demographics of those most negative also reflect what is known of more conspiratorial thinking. For example, the lower educational attainment is consistent with the finding that more conspiracy theory views of COVID are found in those with lower scientific literacy [27] and tend to be right wing (although some are extreme left wing) [28].

We do not discount Dunning–Kruger type effects, not least because such effects may well interact with conspiratorial thinking [5]. Motta and colleagues [5], for example, suggest that overconfidence could lead to dismissal of reliable sources of information about vaccines and be associated with openness to misinformation, both of which characterise conspiratorial thinking. Similarly, prior studies find an influence of educational attainment on susceptibility to fake news [29] and that a lack of relevant knowledge is associated with poor truth discernment [30]. That religiosity appears, before and after control for covariates, as a predictor of the OSD, suggests an additional tension between religiosity and science (see also [31]).

## Caveats

These results come with numerous caveats, the most important of which question the generality of our results. For our survey data, we recover the classical positive correlation between knowledge and attitude (S1 Results) seen in cases where attitude to science is not affected by religion/political factors [3]. Whether our results extend to cases where high educational attainment is associated with both strongly positive and strongly negative attitudes [1,2] is unknown. Indeed, these instances tend to be those where religion/political positions appear to affect attitude to the aligned science, for example, climate change in the case of political polarisation, evolution and big bang in the case of religion [1]. Whether attitude strength in our measure is predicted by strength of subjective understanding in such contexts remains to be seen, although Light and colleagues [6] report no such effect when considering evolution, big bang, and climate change using their scoring system. However, we do identify a tendency for negativity attitudes as regards the Trust and Hype questions to be accentuated in those with right-wing attitudes. It is indeed notable that the trend we discern for these 2 questions is stronger than for the GM question where responses are unaligned with political identity (data here and [1]). Thus, we cannot dismiss the possibility that even some attitudinal questions aligning with political positions can have underpinning self-confidence as the key predictor of attitudinal strength. We suggest that more surveys of our symmetrical form would be of value.

In addition, while we have focused on public attitudes to genetics, not all of our science knowledge questions were biological. Likewise, 2 of our subjective understanding questions concerned science more generally. Might our results be misleading for not focusing more on biological/genetical questions? To address this concern, we have reanalysed the data this time defining "scientific knowledge" employing exclusively the questions that were biological (i.e., omitted those about radioactivity and atoms) and defining subjective understanding by the 4 questions that were explicitly genetical (i.e., questions 3 to 6, see Methods). The main conclusions are unaffected (S2 Results) but we observe 3 qualitative changes to the results (meaning a switch from significance to nonsignificance or vice versa). These are (1) the partial correlation between political attitude and OSD is now marginally nonsignificant when permitting Hype as a covariable (Table 2, lower panel) (it remains marginally significant when Hype is removed as a covariable, Table 3, lower panel). (2) In the context of GM attitudes when we restrict analysis to only instances where the attitude score is less than or equal to zero, the slopes of the 2 lines (subjective or objective versus attitude) are now significantly different, whereas before they were not, although the trend was

in the expected direction (new results: $P = 0.031$, $t = -0.1.86$, df = 2,982; slope subjective = $-0.0353$ +/− 0.0081; slope for objective = $-0.0096$ +/− 0.0112). This strengthens the evidence that those rejecting GM technology have higher subjective understanding than objective knowledge. (3) For the Hartigan and Hartigan dip scores, the distribution of knowledge scores for those most negative in attitude is significantly non-unimodal in all cases except Hype. Previously, GM was unique in this regard. We conclude that our core findings are resilient to, and to some degree strengthened by, restriction of questions that are exclusively biological/genetical.

The results, in addition come with the usual caveat that that for all effects we have no evidence of causality. The questionnaire approach always has potential limitations as regards a biased set of respondents, although the profile of the respondents (S1 Materials) is demographically relatively unbiased.

## Implications

These caveats aside, the results have relevance to PUS on several fronts. First, they question the assumption that distrust in science, and negative attitudes more generally, among those with lower knowledge/education stems from fear of the unknown [3]. Our results suggest that many of these individuals believe that they do understand the science and its consequences: if they fear anything, it is their alternative "known." Similarly, that we see that the same trends when specific technologies are mentioned and when they are not, strongly supports the view that specific technological concerns (the alternative "knowns"), may be for some, a smokescreen concealing a more generalised antipathy. If so, then addressing the specific concerns will do little to overcome the deeper negativity. Follow-up qualitative analysis would be valuable.

Second, these results suggest an alternative approach to tackling science denial/rejection, one that focuses on OSD [4] rather than, or alongside, gain in knowledge/understanding. Training to improve logical reasoning skills helps correct similar OSD [10]. It has been proposed that asking people to explain their understanding of the science may also help, as failure to explain the underlying mechanisms disturbs the sense of understanding [32,33]. The claim [33] that this can lead to moderation of position could not, however, be replicated [34], although other evidence suggests that people are more likely to change their attitudes when their confidence in their knowledge underpinning the attitude is reduced [12]. This strategy also runs the risk of inducing backfire effects [35], whereby individuals become more entrenched when challenged. Whether the focal sceptic changes his or her mind, however, need not be the issue. Instead, prior results [33,34] suggest that explication of understanding, when understanding is weak, leads to reduced confidence in that understanding. Thus, a debating strategy would be to ask those most sceptical of the science to explain their understanding of the science. They commonly will be unable to do so [33,34], which may make their case less persuasive to others. This effect on others is worth experimental investigation.

Third, and related, it suggests a new potential issue. Just as misinformation is more likely to stick if presented repeatedly [36–39], so too if presented with confidence [40,41]. For example, a witness at a crime scene can be convinced by confident presentation of incorrect blame from a co-witness [40]. We have identified a rare set of highly overconfident individuals, who are strong candidates for being disproportionate spreaders of incorrect information. When journalists are seeking "balanced" views on an issue, they should not only source individuals with differing opinions, but also address the factual correctness of their assertions. They should not mistake confidence with competence.

Fourth, the results of this study might be employed as an inoculation against some misinformation [42–44]. If peoples' willingness to accept information from a speaker operates in a Bayesian manner [45], conditioned on priors, then there could be utility in it being known

that those in these most extreme positions are self-confident in their positions but that does not necessarily imply high competence. Recent evidence [44] finds that well-constructed inoculation videos can enable increased resilience to common tactics for spreading misinformation (use of emotionally manipulative language, false dichotomies, incoherence, scapegoating, and personal attacks). It would be valuable to discern whether people could similarly be inoculated against misinformation contagion owing to overconfident delivery.

Fifth, there may, we suggest, be a tendency, at least in the UK context, to focus too much on the extreme science sceptics. Given that overconfidence is associated with lower openness to new information [46] and given the tendency for the most sceptics to not trust anyone (see above), there may be a case to focus more on the majority not this minority. In our surveys, these extreme rejectionists were 1% to 2% of the population (5% for GM, 4% for vaccine—with 2% preferring not to say). In PUS, we should perhaps focus more on the quiet majority than on attempting to convince outliers. Indeed, in our survey, less than 10% of the population said there was too much science coverage while 44% wanted more.

Additionally, by serendipity, we discovered a simple badging exercise that should in principle help. Previously, emphasis has been put on building trust [27], rather than just informing people. Indeed, in some cases where instruction led to increased acceptance, it is not clear that instruction per se was causal [47] while endorsement from trusted figures was [47]. Who the public trust is then an important question. When we asked which sources respondents did trust for information about COVID, a model of familiar or respected individuals seems to predict the trusted: research scientists or Universities were the most trusted (83%), with government's scientific advisors (75%), and NHS spokespersons following (65%) (we note this may not generalise to the US where government science has become politicised). By contrast, not for profit organisations and charities (25%), the government (28%), and work colleagues (5%) all fail to have majority trust. For PUS work University academics should use unambiguous University attachment, not an academic society attachment, presumably because the latter (not for profits) is assumed to be partisan.

## Methods

### The survey

The survey was commissioned through Kantar Public (Kantar Public's random sample panel Public Voice, panel survey 11, June 2021). The survey comprised 1 survey, funded by and for the Genetics Society. The target population was UK individuals aged 18+ and living in residential accommodation. Further details are in S1 Materials including design, implementation (fieldwork), and demographics. Respondents were paid. The data was not collected to specifically test this hypothesis and so was not preregistered. For the source of questions, see S2 Materials.

### Ethics statement

The survey was commissioned by The Genetics Society to be performed by Kantar Public. Kantar Public adheres to the following standards and industry requirements: Market Research Society (MRS) and ESOMAR (the global voice of the data, research, and insights community) professional codes of conduct, ISO 20252 international market research quality standard, ISO 9001 international standard for quality management systems, and the Data Protection Act 2018. Given that respondents had granted consent to Kantar Public to be enrolled on their panel, further ethics approval was not required by Kantar Public for this particular research, but the MRS code of conduct was followed which provides ethical guidelines for the industry. Participants were paid by Kantar Public. The University of Bath ethics committee determined

that with Kantar Public's prior ethical approval, prior consent, and data handling standards, any further ethical approval process was not required and the research was approved on this basis.

## Measurement of subjective understanding

Respondents were asked to consider their understanding in science in general (Q1 and 2) and 4 specific to genetics (Q 3–6):

Q1. Thinking of the stories about science you see or hear in the news, which of the following statements would you say best describes you?

Subjects were then given 4 options: I usually understand what they are talking about; I sometimes understand what they are talking about; I usually do not understand what they are talking about; I do not see or hear science news stories.

These were scored 3 to 0, first to last.

Q2. How well informed do you feel, if at all, about science, and scientific research and developments?

Subjects were then given 4 options: Very well informed; Fairly well informed; Not very well informed; Not at all informed. These were scored 3 to 0, first to last.

Q3. First, when you hear the term DNA, how would you rate your understanding of what the term means?

Q4. Next, when you hear the term GM or genetically modified, how would you rate your understanding of what the term means?

Q5. Next, when you hear the term natural selection, how would you rate your understanding of what the term means?

Q6. Next, when you hear the term PCR, how would you rate your understanding of what the term means?

These 4 questions invited responses: Very good; Good; Some understanding; Have heard the term but have little understanding of what it means; Have not heard the term. Questions 3 to 6 were scored from 4 to 0, first to last.

All values were normalised by dividing by the maximum score for any given question, the single self-assessment score being the mean of these 6 normalised responses. These scores therefore range from 0 (no self-confidence) to 1 (perfect self-confidence). The 6 questions have acceptable to good consistency (Cronbach's alpha = 0.79 CI 0.775 to 0.802, $N$ = 2,065).

## Assessment of scientific knowledge

Respondents were asked 12 true-false questions, mostly drawn from prior surveys but with the addition of some current to the COVID pandemic:

(1) All plants and animals have DNA; (2) the oxygen we breathe comes from plants; (3) the cloning of living things produces genetically identical copies; (4) by eating a genetically modified fruit, a person's genes could also become modified; (5) all radioactivity is human-made; (6) it is the mother's genes that determine the sex of the child; (7) electrons are smaller than atoms; (8) tomatoes do not naturally contain genes—genes are only found in genetically modified tomatoes; (9) dinosaurs and humans share a common ancestor; (10) the spread of new variants of viruses can occur through natural selection; (11) COVID-19 is caused by bacteria; and (12) viruses are smaller than bacteria.

Answers were scored as +1 correct or −1 incorrect and each respondent given a science/genetics knowledge score from −1 to +1 taking the mean. The 12 questions have "poor" internal consistency (Cronbach's alpha = 0.56 95% limits 0.532 to 0.583, $N$ = 2,065). No question

was particularly influential (alpha values when each question dropped range from 0.5 (Q5, radioactivity) to 0.58 (Q2, oxygen from plants)).

## The OSD score

On our metrics, a score of 1 for the knowledge test means all answers were correct, while a score of 0 is equivalent to guessing all answers (the questions were true/false and answers were required). Similarly, on the subjective assessment scale, a score of 0 indicates no confidence in own ability and a score of 1 indicates full confidence. Given this, if we assume that those with a knowledge score of 1 are full experts, then the expected absolute value of the self-assessment score, if self-assessment is a perfectly accurate reflection of knowledge, would be the objective knowledge test score (i.e., a line of slope one running through the origin in Fig 2 would describe perfect self-understanding: light green line). Distance above or below this line thus defines our OSD score, i.e.:

OSD = knowledge score—self-assessment score.

We also provide an alternative scoring system, $Z_{OSD}$. We define:

$Z_{know}$ = (knowledge score–mean of all knowledge scores)/SD of all knowledge scores

and

$Z_{subj}$ = (self-assessment score–mean of all self-assessment scores)/SD of all self-assessment scores,

then

$Z_{OSD} = Z_{know} — Z_{subj}.$

The absolute scale has the advantage that, owing to our scoring system, the absolute value of OSD is indicate of the absolute degree of over or under confidence, perfect alignment sitting on the line of slope 1 running through the origin. The Z-based system, by contrast, provides a measure of overconfidence that is relative to others in the sample but is not absolute.

## Estimation of covariates

We considered 4 covariates: age, educational attainment, religiosity, and political identity. Age was calendar age in years at the time of the survey. Educational attainment was classified according to respondent selected answer as to whether they were "Not religious," scored 0, "Religious (not actively practicing)," scored 1, and "Religious (practicing)," scored 2. Educational attainment was scored: "Degree level qualification(s)" = 2, "Non-degree level qualifications" = 1, "No academic or vocational qualifications" = 0.

To assay political identity, subjects were asked 10 questions (not presented in this order):

1. Rich people can get away with breaking the law.

2. Peoples working conditions and wages need strong legal protection.

3. Major public services and industries should be in state hands.

4. People in Britain should be more tolerant of those who lead unconventional lives.

5. The government should redistribute income from the better-off to those who are less well off.

6. The monarchy should be abolished.

7. People today do not have enough respect for traditional British values.

8. Business in this country is overregulated by the government.

9. People who break the law should be given tougher sentences.

10. There should be fewer immigrants in this country.

Respondents were invited to say whether they "Strongly agree," "Tend to agree," "Neither agree nor disagree," "Tend to disagree," "Strongly disagree," or "Do not know." Failure to respond was, like "Do not know," classified as NA. Responses were scored −2 to +2 with a more positive response being the more right wing in inclination. For questions 1 to 6, therefore, agreement was scored as negative, and disagreement score positive. Questions 7 to 10 were scored in the inverse manner. A mean score across the 10 questions was calculated for each individual (with NAs eliminated from calculation of the mean). This in turn was divided in 2 to provide our "political identity" scale from −1 (strongly left wing) to +1 (strongly right wing). Cronbach's alpha for the 10 political questions 0.761 (95% CI 0.745 to 0.778, $N = 2,065$), i.e., acceptable to good. To ascertain the relationship between our metric and real-world behaviour, we analysed voting pattern from the 2019 UK General Election. Analysis of the political scores as a function of party voted for supports the metric in so much as right wing parties (DUP, Conservative, Brexit) have positive scores, more left leaning parties (labour, SNP, SDLP) are more negative and centrist parties (Liberal Democrats) are more centrally scoring (see S3 Fig).

## Statistics

All statistics were done in R (v 4.1.0). Violin plots were constructed using package vioplot. Partial correlation analyses employed package ppcor. Correlation matrixes were derived using the rcorr package. The analysis script (including specification of all required libraries) and input data are available from doi: 10.5281/zenodo.7289133. Test of difference in slope on lines was done with a *t* test where:

$$t = (\text{Slope 1} - \text{Slope 2}) / \sqrt{(\text{SEM1}^2 + \text{SEM2}^2)}.$$

*P* is derived from the t distribution assuming a two-tailed test and n1 + n2-4 degrees of freedom. For further details of packages employed, see scripts at 10.5281/zenodo.7289133.

## Supporting information

**S1 Fig.** Attitude to GM (a) as a histogram plot, (b) in relation to educational attainment, and (c) in relation to political attitude. All scripts and data are available at doi: 10.5281/zenodo.7289133.
(PDF)

**S2 Fig.** Vaccine attitude (a) as a histogram plot, (b) as related to sex ratio, (c) in relation to educational attainment, and (d) in relation to political attitude. All scripts and data are available at doi: 10.5281/zenodo.7289133.
(PDF)

**S3 Fig. The relationship between voting pattern in the 2019 UK general election and the political identify score.** All scripts and data are available at doi: 10.5281/zenodo.7289133.
(PDF)

**S1 Materials. The survey contents, demographics, and fieldwork.** All scripts and data are available at doi: 10.5281/zenodo.7289133.
(PDF)

**S2 Materials. The derivation of questions.** All scripts and data are available at doi: 10.5281/zenodo.7289133.
(PDF)

**S1 Table. Correlations between attitudinal scores and the 12 knowledge questions.** All scripts and data are available at doi: 10.5281/zenodo.7289133.
(XLSX)

**S2 Table. Tukey post hoc test results for 3 parameters (self-assessed understanding, knowledge, and OSD) as a function of vaccine attitude.** All scripts and data are available at doi: 10.5281/zenodo.7289133.
(PDF)

**S3 Table. Hartigan and Hartigan dip tests on scientific knowledge scores for those most sceptical on various measures.** For Trust, Hype, and GM, these are class −2. For vaccine, it is those who will not have it. All scripts and data are available at doi: 10.5281/zenodo.7289133.
(PDF)

**S4 Table. Correlations between all 12 knowledge score answers.** A high correlation implies those who answered 1 question correctly[incorrectly] also answered the other one correctly [incorrectly]. All scripts and data are available at doi: 10.5281/zenodo.7289133.
(PDF)

**S1 Results. The data accords with expectations of unpolarised science.** All scripts and data are available at doi: 10.5281/zenodo.7289133.
(PDF)

**S2 Results. Replication of all results when knowledge is defined by 10 biology questions and subjective understanding is defined by the 4 genetic questions.** All scripts and data are available at doi: 10.5281/zenodo.7289133. In this instance, consult zip archive S_results_2.zip.
(PDF)

## Acknowledgments

We thank Patrick Sturgis and Sarah Cunningham-Burley for advice on survey design and on the manuscript.

## Author Contributions

**Conceptualization:** Jonathan Pettitt, Alison Woollard, Wendy Bickmore, Laurence D. Hurst.

**Data curation:** Cristina Fonseca, Laurence D. Hurst.

**Formal analysis:** Laurence D. Hurst.

**Funding acquisition:** Jonathan Pettitt, Alison Woollard, Laurence D. Hurst.

**Investigation:** Laurence D. Hurst.

**Methodology:** Cristina Fonseca, Jonathan Pettitt, Alison Woollard, Adam Rutherford, Wendy Bickmore, Anne Ferguson-Smith, Laurence D. Hurst.

**Project administration:** Cristina Fonseca.

**Software:** Laurence D. Hurst.

**Visualization:** Laurence D. Hurst.

**Writing – original draft:** Laurence D. Hurst.

**Writing – review & editing:** Jonathan Pettitt, Alison Woollard, Adam Rutherford, Wendy Bickmore, Anne Ferguson-Smith, Laurence D. Hurst.

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
