## [Editor Report · Decision Letter 0]

20 Jul 2022

Dear Laurence, 

Thank you for submitting your manuscript entitled "Why people have different attitudes to science: it isn’t just about what you know but what you think you know" for consideration as a Research Article by PLOS Biology. Please accept my apologies for the delay in getting in touch with a decision on your work. Roli is the handling editor assigned to the submission, but went on a much deserved holiday on Friday without having had time to read your submission, so I have taken over handling it during his absence.

I have now had time to read your manuscript and discuss it with an academic editor of relevant expertise and I am writing to let you know that we would like to send your submission out for external peer review. 

However, before we can send your manuscript to reviewers, we need you to complete your submission by providing the metadata that is required for full assessment. To this end, please login to Editorial Manager where you will find the paper in the 'Submissions Needing Revisions' folder on your homepage. Please click 'Revise Submission' from the Action Links and complete all additional questions in the submission questionnaire. Our academic editor remarked that they would like to know much more about how they derived the survey questions. The methods state that "most are from previous surveys," but no references are provided. As this is an question that the reviewers may also raise, you might want to modify your manuscript file to include this information at this time. Please note, however, that we expect to receive your full submission in 1-2 days. Please let us know if adding this information is likely to take much longer, and if so perhaps it'd make sense to do this at a later stage. 

Once your full submission is complete, your paper will undergo a series of checks in preparation for peer review. After your manuscript has passed the checks it will be sent out for review. To provide the metadata for your submission, please Login to Editorial Manager (https://www.editorialmanager.com/pbiology) within two working days, i.e. by Jul 22 2022 11:59PM.

Kind regards,

Nonia

Nonia Pariente, PhD

Editor in Chief

PLOS Biology

npariente@plos.org

on behalf of

Roli

Roland Roberts, PhD

Senior Editor

PLOS Biology

rroberts@plos.org

---

## [Decision Letter · Decision Letter 1]

16 Sep 2022

Dear Laurence,

Thank you for your patience while your manuscript "Why people have different attitudes to science: it isn’t just about what you know but what you think you know" was peer-reviewed at PLOS Biology. It has now been evaluated by the PLOS Biology editors, an Academic Editor with relevant expertise, and by three independent reviewers. 

In light of the reviews, which you will find at the end of this email, we would like to invite you to revise the work to thoroughly address the reviewers' reports.

You'll see that the reviewers are broadly positive about your study, but reviewers #1 and #2 each raise a number of concerns that should be addressed. After discussing the reviews with the Academic Editor, we feel that there is clear value in your paper beyond the studies mentioned by the reviewers. I should also point out that the Light et al. paper mentioned by one of the reviewers is amply covered by the six-month term of our anti-scooping policy (and indeed may even have been published after your paper was submitted); nevertheless, we encourage you to cite and discuss it.

Given the extent of revision needed, we cannot make a decision about publication until we have seen the revised manuscript and your response to the reviewers' comments. Your revised manuscript is likely to be sent for further evaluation by all or a subset of the reviewers.

**IMPORTANT - SUBMITTING YOUR REVISION**

*Re-submission Checklist*

*Published Peer Review*

*PLOS Data Policy*

*Blot and Gel Data Policy*

Best wishes,

Roli

Roland Roberts, PhD

Senior Editor

PLOS Biology

rroberts@plos.org

REVIEWERS' COMMENTS:

Reviewer #1:

In the current manuscript, the authors ask if UK individuals' subjective understanding of science and related topics predicts their attitudes on those topics and technologies. They present data supporting this idea, finding that people have higher levels of subjective knowledge when they have more extreme attitudes, be they supportive or oppositional. They aim to make a contribution by pointing out the positive correlation between subjective knowledge and attitudes at both ends of the attitudinal distribution, and by replicating these relationships when the attitude object is genetic science and scientists more generally, instead of specific genetic technologies. 

First, I would like to commend the authors on a clear, well-written, well-evidenced paper. The topic is timely and the empirical findings are of interest to both scientists and laypeople, both inside and outside of the biological sciences.

I have one main critique, which I discuss first. Then I will make several comments to elaborate on the main critique, all of which are numbered point 1, point 2, etc. for easy tracking. Beyond the main critique, the order of comments should not be taken to indicate degree of importance. I conclude the review with how I believe the manuscript could be improved.

The main critique is that the magnitude of the contribution is a bit stretched. The idea of testing whether "extremists" on both the positive and negative sides of the attitude spectrum are more confident in their understanding is important, since much of the prior research on the topic has focused on people whose attitudes oppose those supported by many scientists. This is the biggest contribution of the paper, in my view. However, this contribution is weakened by a few things. The first (point 1) is that although this paper aims to extend the work of Fernbach et al. (2019) and others by examining the subjective understanding-attitude relationship on a bipolar subjective knowledge scale, the scale itself has only five points (fewer than Fernbach et al.'s seven). It is true that two of these scale points go beyond the most extreme level of agreement in the Fernbach et al. paper, but the J shape of the relationship is nearly identical to those reported in the Fernbach paper, as well as in a recent paper in Science Advances (Light et a. 2022) (point 2), which suggests that participants may be distributing similarly across the two scales in the three papers. The fact that the trend is more of a J shape than a true U shape (which is rightly pointed out by the authors in their individual tests of the slopes on both sides of neutral) also slightly weakens the overall claim relying on subjective knowledge to predict attitude extremity, since the relationship is stronger for those who hold opposition attitudes, and opponents are more problematic (societally, scientifically, etc.) than those with attitudes more in line with science (point 3). Here is what is written in Light et al. 2022: 

"The subjective knowledge distribution is j-shaped (see the Supplementary Materials for visualizations). Similar to more extreme opponents, those fully in line with the scientific consensus rated subjective knowledge higher than moderate opposers (but lower than extreme opponents). However, whereas the confidence of those in agreement with the established science is substantiated by their actual knowledge, the confidence of extremists appears to be misplaced."

A small additional point here is that there appear to be very few respondents at the extreme agreement end of the attitude distributions (point 4). In other words, while useful and interesting, the absolute value approach may obscure the result that most of the action appears to be happening on the opposing side of the scale.

Another reason that I believe the magnitude of the contribution to be a bit stretched is that the authors do not discuss the scientific topics both in Fernbach et al. 2019 and Light et al. 2022 where the relationship between subjective understanding and attitudes was near-zero or not statistically significant. They were climate change, the Big Bang, and Evolution (point 5). It is possible that the authors of the current manuscript intended the paper's generalizations regarding subjective understanding to apply only to a specific subset of scientific issues relating to genetics, in which case this critique may be misplaced and the authors are justified in not discussing those findings, but this could be helped by adding some clarifying writing on the point (and perhaps editing the manuscript title as well).

I was also a little confused when the authors discuss the implications of the Hype and Trust analyses. Because these items focused on genetic science and scientists more broadly, the authors interpret finding the same pattern of results as the more specific measures as evidence that objections based on specific technologies may be "a smokescreen hiding a more general antipathy." This may or may not be true, depending on what the authors meant. It is my hunch that the Hype and Trust measures are actually capturing more extreme participants—being broadly anti-science is more extreme than being anti-GM (point 6). So, someone who disagrees with those may not act the same when asked about a specific technology, which they may not evaluate in the context of where the technology sits among their larger anti-scientific attitudes or identity. This is speculation on my part, but it's also an empirical question that the authors could test. So, while it is true that people who are broadly anti-science are likely to also be against specific technologies they view as "scientific," this does not mean that all those against a specific scientific technology are broadly anti-scientific, which is how the current Hype and Trust implications read to me. It's an important distinction—have you uncovered a broad psychological phenomenon or another specific instantiation of the phenomenon among a more extreme population?

I do not believe it is the role of reviewers to decide whether a manuscript is suitable for publication in a specific journal (that is the job of the editors), so I will instead provide some thoughts on how I believe the paper could be improved.

a) Focusing on the bipolar nature of the subjective understanding - attitude relationship.

b) Focusing less on the manuscript's current replications of prior work, and being clear about what is new.

c) Doing so in new studies with larger, nationally-representative samples from the UK or elsewhere in order to make sure there are enough participants at the extremes to have confidence in the bipolar findings. Also including additional scientific topics if the authors intended for their findings to be more broadly generalizable.

d) Editing the writing to be more specific so as to ensure that the magnitude of the contribution and intended generalizability of the findings does not come off as overblown. 

e) Providing histograms of the distributions of participants over attitude levels for all studies. This is a smaller ask, of course, but I could only find them for the Trust and Hype measures.

Best of luck to the authors, and thank you for working on this important topic.

Reviewer #2:

This is an interesting paper on an important issue, one I have struggled with in more than one study. The key question of the paper is whether attitudinal strength is predicted by subjective understanding (p. 5 of the manuscript); the fact that it is in the domain of genetics is an important result. To my mind, the many subquestions the authors pose are informative, and the statistical tests used to examine them seem valid. My primary complaint is that the methods are opaque enough that assessment of the work is difficult and competing explanations cannot be ruled out. I also urge caution in the concluding recommendations.

(1) The dependent variable is described as the "modulus" of the Hype and Trust scales (p. 5). I was not sure what this means and asked a mathematician, who agreed that the authors probably meant "absolute value." The R code confirms this, but this critical detail should be stated outright in the manuscript.

(2) One independent variable is described as the mean of six normalized (z-scored?) scales meant to capture subjective understanding. For two of the six questions, I couldn't even hazard a guess as to the response scales—the manuscript only says "Questions 1 and 2 invited a response in one of four ways and were graded 1-4 (4 maximum)." What are these ways? The response options are in the supplemental materials, but for a key variable, this information should not be buried so deeply. Moreover, even once I dug them up, it was not clear how the questions were in fact coded. For the "well-informed" question (Q1112), I assume responses were reverse coded. If that's right, the manuscript should say so. For the "understanding-science-in-the-news question" (Q1111), I really can't say—were they reverse coded, or reverse-coded omitting option 4 "I don't see or hear science news stories," or something else? I could not find this information in the R code; either I missed it, or this step occurred prior to the analysis code.

Of course, these details probably do not affect the big picture, since the measure used in the analyses is a mean over all scales, and these scales showed good internal consistency. But the lack of clarity on these details will confuse the reader trying to follow the logic of the study. 

(3) It is never explicitly stated why the authors chose the covariates that they did. It wasn't clear to me from the supplementary materials whether the Public Voice survey measured other demographic characteristics, but political ideology is a conspicuous possible confound. If age, religiosity and education were the only available covariates, the manuscript should say so; if they were chosen for particular reasons, the manuscript should state them. 

(4) The other independent variable is clear enough in how objective knowledge is operationalized. But it raises another quite general, conceptual problem with the measures used, one that some of the papers the authors cite also struggle with. Call it the domain congruence problem.

In Replication 1 (p. 13), the authors report the same relationship between the objective/subjective deficit and attitudes towards genetically modified foods that were reported in Fernbach, Motta, etc. The trouble is that the OSD is the difference between an objective knowledge score and a subjective knowledge score, neither of which exclusively concerns genetics. It is true that the objective knowledge measure contained a number of basic science facts that sit squarely in the domain of genetics, but it contained plenty that arguably (viruses, dinosaurs) or inarguably (electrons, radioactivity) do not. The low internal consistency of these scales, while unsurprising for an objective knowledge measure, is surely an opening for the critic of the authors' approach. How is the reader to know that the main results are not an artifact of averaging over all of these measures, only some of which have to do with genetics? In other words, rather than people who most oppose genetics knowing relatively little about genetics, it could be that people who most oppose genetics know relatively little about science-y stuff. 

The subjective knowledge measure has the same problem—some questions are straightforwardly about genetics, others are about how well people feel they understand science news—although in that case, internal consistency was high.

If, for example, it turned out that (a) the effects the authors report are being driven entirely by the subjective knowledge questions about science news and (b) the response option "I don't see or hear science news stories" was included as the endpoint of the "understanding-science-in-the-news" question, meaning the lowest degree of subjective understanding is assigned to people who don't pay attention to science news at all, then the main analyses do not reflect a relationship between subjective understanding and attitudes, but rather between consumption of mainstream science news and attitudes. This is a quite different phenomenon than the one the paper is meant to address, and the brief note in the Discussion about trust in media sources further supports this alternative explanation. The same problem applies to COVID subjective and objective knowledge, of course.

One way to address this problem is to test for the same patterns in the data using only the subscales that are unequivocally about genetics (or COVID). This may seem like a chore, but I raise the issue because Light and colleagues have been understood as claiming that dumb people oppose science and smart people support it. I believe the picture is more complicated than this, and the current paper deepens our understanding of the complex interplay between attitudes, subjective knowledge, and objective knowledge. But if the authors wish to avoid also being understood as saying that dumb people oppose science, they should address the domain congruence issue with further analysis or additional discussion. 

(5) With full recognition that specifying the practical implications of these findings is extremely difficult, I submit that "asking people to explain their understanding of the science" (p. 20) will almost certainly not help. I work with a number of practitioners in public health and health communication, and I have yet to find one that believes making others feel stupid is a good approach to softening attitudes toward policies or interventions that the best science suggests are safe and effective.

Moreover, "friends and family" and "government's scientific advisors" are anything but non-partisan sources, albeit for different reasons. For friends and family, there is damning evidence of homophily (Mosleh et al., 2021, "Shared partisanship dramatically increases social tie formation in a Twitter field experiment"), meaning trusting one's friends and family hardly precludes ideological sorting. Perhaps scientific advisors to the government are still considered neutral in the UK, but they are not in the US; in a recent, qualitative measure probing people's reasons for trusting or not trusting information from public health scientists, those who do not trust it invoked our top COVID advisor Anthony Fauci almost constantly despite the fact that he has not appeared on TV in roughly a year.

What should the authors recommend based on their findings, in that case? I wish I knew, but at the very least I would urge further reflection on this important question. 

Reviewer #3:

Why people have different attitudes to science: it isn't just about what you know but what you think you know

Cristina Fonseca, Jonathan Pettitt, Alison Woollard, Adam Rutherford, Wendy Bickmore, Anne Ferguson-Smith , Laurence D. Hurst

The abstract is good, and explains why the studies are important and of general interest.

In general, I think this paper makes a definite contribution to the field, in terms of examination of both positive and negative attitudinal extremes, and examination of the generality vs specificity of the question focus. The analysis seems quite sophisticated. I found it very dense, with both the analysis and the style often hard to follow. This may be more a function of my limitations, rather than a problem with the paper.

---

## [Decision Letter · Decision Letter 2]

2 Nov 2022

Dear Laurence,

Thank you for your patience while we considered your revised manuscript "Why people have different attitudes to science: more extreme attitudes are associated with high self-confidence, even if this isn’t justified" for publication as a Research Article at PLOS Biology. This revised version of your manuscript has been evaluated by the PLOS Biology editors, the Academic Editor, and two of the original reviewers.

Based on the reviews, we are likely to accept this manuscript for publication, provided you satisfactorily address the remaining points raised by the reviewers. Please also make sure to address the following data and other policy-related requests.

IMPORTANT:

a) I discussed reviewer #1's remaining concerns with the Academic Editor, who felt, like me, that you should be left to read their comments and revise your manuscript as you see fit. We see this as an ongoing debate, and do not require new data or analysis.

b) Please change your Title to something that avoids punctuation; we suggest something like "More extreme attitudes to science are associated with high self-confidence even when unjustified" or "More extreme attitudes to science are associated with unjustified high levels of self-confidence" (I concede that these two are not synonymous).

c) Thank you for the detailed data and code provision in Github. However, because this can be changed at any time, we need an immutable DOI'd version of record to be provided, for example in Zenodo.

d) Please cite the location of the data clearly in all relevant main and supplementary Figure legends, e.g. “The data underlying this Figure can be found in https://doi.org/XXXX”

We expect to receive your revised manuscript within two weeks. 

*Published Peer Review History*

*Press*

Best wishes,

Roli

Roland Roberts, PhD

Senior Editor,

rroberts@plos.org,

PLOS Biology

DATA NOT SHOWN?

Reviewer #1:

[IMPORTANT: See attached document for substantially formatted review]

Reviewer #2:

I commend the authors for diligently responding to all of my concerns, and more generally for grappling sincerely with quite subtle issues. Please note one error in the latest revisions: The authors added to the manuscript an explanation of the covariates chosen that concludes, "We have no score of political extremism." This should say "political identity" or "political ideology." Extremism is another matter entirely!

---

## [Editor Report · Decision Letter 3]

14 Nov 2022

Dear Laurence,

Thank you for the submission of your revised Research Article "People with more extreme attitudes towards science have self-confidence in their understanding of science, even if this isn’t justified" for publication in PLOS Biology. On behalf of my colleagues and the Academic Editor, Lisa Bero, I'm pleased to say that we can in principle accept your manuscript for publication, provided you address any remaining formatting and reporting issues. These will be detailed in an email you should receive within 2-3 business days from our colleagues in the journal operations team; no action is required from you until then. Please note that we will not be able to formally accept your manuscript and schedule it for publication until you have completed any requested changes.

Sincerely, 

Roli

Senior Editor

PLOS Biology

rroberts@plos.org